# Real-Time Speed Estimation for an Induction Motor: An Automated Tuning of an Extended Kalman Filter Using Voltage–Current Sensors

**DOI:** 10.3390/s24061744

**Published:** 2024-03-07

**Authors:** Ines Miloud, Sebastien Cauet, Erik Etien, Jack P. Salameh, Alexandre Ungerer

**Affiliations:** 1Université de Poitiers, ISAE-ENSMA Poitiers, LIAS, 86073 Poitiers, Franceerik.etien@univ-poitiers.fr (E.E.); 2Chauvin Arnoux, 74940 Annecy, France; jack.salameh@chauvin-arnoux.com (J.P.S.); alex.ungerer@chauvin-arnoux.com (A.U.)

**Keywords:** Kalman filter, noise covariance matrix, subspace model identification, induction motor

## Abstract

This paper aims at achieving real-time optimal speed estimation for an induction motor using the Extended Kalman filter (EKF). Speed estimation is essential for fault diagnosis in Motor Current Signature Analysis (MCSA). The estimation accuracy is obtained by exploring the noise covariance matrices estimation of the EKF algorithm. The noise covariance matrices are determined using a modified subspace model identification approach. In order to reach this goal, this method compares an estimated model of a deterministic system, derived from available input–output datasets (using voltage–current sensors), with the discrete-time state-space representation used in the Kalman filter equations. This comparison leads to the determination of model uncertainties, which are subsequently represented as noise covariance matrices. Based on the fifth-order nonlinear model of the induction motor, the rotor speed is estimated with the optimized EKF algorithm, and the algorithm is tested experimentally.

## 1. Introduction

The induction motor (IM) is by far the most widely used motor in industry [1], as this motor is found in furnaces, conveyors, pumps, and other industrial equipment. Monitoring and controlling induction motor speed is essential because changes or abnormalities in motor speed can indicate potential faults or malfunctions. Furthermore, speed estimation is crucial for fault diagnosis of the IM. Unfortunately, it can be challenging or extremely expensive to use sensors for the speed measurement of IM. In fact, some industrial environments may have unsuitable conditions for sensor implementation such as high temperatures, vibrations, or corrosive substances. To overcome these challenges, alternative methods can be employed for the real-time speed estimation of IM. One commonly used technique consists in estimating the speed from the motor electrical variables only using voltage–current sensors. This is typically achieved using algorithms or observers such as Sliding Mode Observers [2], Model Reference Adaptive Systems [3], or the Extended Kalman Filter [4,5,6,7,8,9]. Among all these methods, the EKF method has attracted significant attention due to its good estimation accuracy in the presence of noise. In the literature, there are different studies in which EKF based state/parameter estimation is executed. Those studies can be divided into two groups. The first group is where EKF is applied, and it is considered that no mechanical information is available [4,5,6]. The second group of solutions makes use of the rotational speed through the equations of motion [7,8,9]. The latter requires knowledge of the mechanical parameters such as load torque, inertia varying with the load coupled to the shaft, and viscous friction. In this paper, we focus on the first group of studies assuming the lack of information about the mechanical parameters.

One key feature of the Kalman filter is the requirement to process and measure the noise covariance matrices. These matrices represent the noise levels in the system and have a direct impact on filter performance. In fact, incorrect or badly adjusted values of *Q* and *R* can lead to inaccurate estimation of the system state and filter instability. Furthermore, the parameters of the noise covariance matrices *Q* and *R* are usually determined by trial-and-error procedures, which may not give accurate results. As the distribution of noise is usually unknown, it is not possible to deduce a generic relationship between the values of the matrix elements and the EKF performance to yield the best speed estimation results.

The determination of noise covariance matrices is still an open issue in practice. A significant part of the literature dedicated to Kalman filter tuning is devoted to the development of techniques and algorithms for the determination of these covariance matrices. However, there are a limited number of studies using covariance matrices estimation for the real-time speed estimation of IM. In fact, researchers focus on investigating the optimization of noise covariance matrices for induction motor speed estimation. This investigation can be categorized into two main parts: the first group of studies [10,11] uses artificial intelligence-based methods, which requires expert knowledge and involves a complex design procedure, whereas the second group [12,13,14,15,16,17,18,19] employs adaptive structures to eliminate the adverse effect under operating condition variations.

The main contribution of this study is to design a fifth-order IM model, where the IM speed is included in the model as a state without the help of the equation of motion, and the noise covariance matrices are estimated. Among all of the Kalman filter tuning techniques available in the literature, specific attention is paid herein to the noise covariance matrices’ estimation with the modified subspace model identification method [20]. The method consists in translating the discrepancy between the identified model (determined from the available input–output datasets) and the discrete-time state-space representation involved in the Kalman filter equations into noise covariance matrices estimates. The model identification method used in [20] was derived from the subspace model identification methods [21,22,23,24] and adapted to estimate a discrete linear time invariant state space model. The main motivation for selecting this class of methods is the strong and inherent link between the subspace identification methods and Kalman filtering [25].

In this paper, a noise covariance matrices estimation with a modified subspace model identification method is employed in order to achieve the best EKF performance in real-time speed estimation. In order to reach this goal, the paper is organized as follows: Section 2 presents the dynamic model of the induction motor and the extended dynamic model of the IM to be used in the EKF. Section 3 details the design of the EKF to estimate the motor speed. Section 4 presents the noise covariance matrices estimation procedure by using a modified subspace model identification method. Section 5 is devoted to the application of the modified subspace model identification method to rotor speed estimation. Section 6 compares the experimental performance of the trial-and-error-process method and the modified subspace model identification method used to determine the noise covariance matrices of the EKF in the case of rotor speed estimation. Finally, Section 7 gives the conclusions.

## 2. Dynamic Model of Induction Motor

In order to estimate the real-time motor speed using EKF, the mathematical model of the three-phase induction motor is developed. Because three-phase systems involve complex quantities and calculations, the analysis of the three-phase induction motor is simplified using the Clarke transformation. In this way, the three-phase quantities (*a*, *b*, *c*) are transformed into two-phase quantities (α, β), which are aligned with the stator (stationary reference frame) [26].

Figure 1 shows the equivalent circuit of the IM in the stationary reference frame [27].

In this equivalent circuit, $R_{s}$ and $L_{1s}$ are the stator resistance and inductance, respectively; $R_{r}$ and $L_{1r}$ are the rotor resistance and inductance referring to the stator side, respectively; $L_{m}$ is the mutual inductance; $\omega_{r}$ is the rotor speed; $v_{s\alpha }$, $v_{s\beta }$ are the measured stator voltages, and $i_{s\alpha }$, $i_{s\beta }$ are the stator currents in the stationary reference frame; $v_{r\alpha }$, $v_{r\beta }$ are the rotor voltages, and $i_{r\alpha }$, $i_{r\beta }$ are the rotor currents in the stationary reference frame; $\lambda_{s\alpha }\left(t\right)$, $\lambda_{s\beta }\left(t\right)$ are the stator flux, and and $\lambda_{r\alpha }\left(t\right)$, $\lambda_{r\beta }\left(t\right)$ are the rotor flux in the stationary reference frame.

Figure 1 shows the electrical and magnetic energy exchanges present in the induction motor, which allow us to derive the mathematical equations.

First, let us introduce a state-space representation of IM dynamics, assuming that the IM speed is a known parameter
(1a)$$\dot{x}\left(t\right)=Ax\left(t\right)+Bu\left(t\right),$$
(1b)y(t)=Cx(t),
where x(t) stands for the state vector, y(t) is the output vector, and u(t) is the input vector, with
(2)$$x\left(t\right)={\left[i_{s\alpha }\left(t\right)\;i_{s\beta }\left(t\right)\;\lambda_{r\alpha }\left(t\right)\;\lambda_{r\beta }\left(t\right)\right]}^{T},$$
(3)$$y\left(t\right)={\left[i_{s\alpha }\left(t\right)\;i_{s\beta }\left(t\right)\right]}^{T},$$
(4)$$u\left(t\right)={\left[v_{s\alpha }\left(t\right)\;v_{s\beta }\left(t\right)\right]}^{T},$$
(5)$$A=\left[\begin{array}{cccc} -\frac{K_{r}}{K_{l}} & 0 & \frac{L_{m}R_{r}}{L_{r}^{2}K_{l}} & \frac{pL_{m}\omega_{r}}{2L_{r}K_{l}}\\ 0 & -\frac{K_{r}}{K_{l}} & -\frac{pL_{m}\omega_{r}}{2L_{r}K_{l}} & \frac{L_{m}R_{r}}{L_{r}^{2}K_{l}}\\ \frac{L_{m}}{\tau_{r}} & 0 & -\frac{1}{\tau_{r}} & -\frac{p}{2}\omega_{r}\\ 0 & \frac{L_{m}}{\tau_{r}} & \frac{p}{2}\omega_{r} & -\frac{1}{\tau_{r}} \end{array}\right],$$
(6)$$B=\left[\begin{array}{cc} \frac{1}{K_{l}} & 0\\ 0 & \frac{1}{K_{l}}\\ 0 & 0\\ 0 & 0 \end{array}\right],$$
(7)$$C=\left[\begin{array}{cccc} 1 & 0 & 0 & 0\\ 0 & 1 & 0 & 0 \end{array}\right],$$
where *p* is the number of poles, $\tau_{r}=L_{r}/R_{r}$ is the rotor time constant, $K_{r}=R_{r}+{L_{m}}^{2}R_{r}/{L_{r}}^{2}$, and $K_{l}=\left(1-{L_{m}}^{2}/L_{r}/L_{s}\right)L_{s}$, with $L_{s}=L_{1}s+L_{m}$ and $L_{r}=L_{1}r+L_{m}$.

As mentioned above, the rotor speed $\omega_{r}$ in Equations ([Disp-formula FD1a-sensors-24-01744]) and ([Disp-formula FD1b-sensors-24-01744]) is considered as a known parameter. If the speed measurement is not available, the rotor speed has to be estimated. To this end, an extended induction motor model including the rotor speed as state variable is developed [4,5]. The rotor speed is then considered both as a state and a parameter, leading to a nonlinear IM model defined as
(8a)$$\dot{x_{e}}\left(t\right)=f_{e}\left(x_{e}\left(t\right),u\left(t\right)\right),$$
(8b)$$y\left(t\right)=C_{e}x_{e}\left(t\right),$$
where $f_{e}$ is the nonlinear function of the states and inputs defined as
(9)$$f_{e}\left(x_{e}\left(t\right),u\left(t\right)\right)=A_{e}x_{e}\left(t\right)+B_{e}u\left(t\right),$$
with
(10)$$x_{e}\left(t\right)={\left[i_{s\alpha }\left(t\right)\;i_{s\beta }\left(t\right)\;\lambda_{r\alpha }\left(t\right)\;\lambda_{r\beta }\left(t\right)\;\omega_{r}\left(t\right)\right]}^{T},$$
(11)$$u\left(t\right)={\left[v_{s\alpha }\left(t\right)\;v_{s\beta }\left(t\right)\right]}^{T},$$
(12)$$A_{e}=\left[\begin{array}{ccccc} -\frac{K_{r}}{K_{l}} & 0 & \frac{L_{m}R_{r}}{L_{r}^{2}K_{l}} & \frac{pL_{m}\omega_{r}\left(t\right)}{2L_{r}K_{l}} & 0\\ 0 & -\frac{K_{r}}{K_{l}} & -\frac{pL_{m}\omega_{r}\left(t\right)}{2L_{r}K_{l}} & \frac{L_{m}R_{r}}{L_{r}^{2}K_{l}} & 0\\ \frac{L_{m}}{\tau_{r}} & 0 & -\frac{1}{\tau_{r}} & -\frac{p}{2}\omega_{r}\left(t\right) & 0\\ 0 & \frac{L_{m}}{\tau_{r}} & \frac{p}{2}\omega_{r}\left(t\right) & -\frac{1}{\tau_{r}} & 0\\ 0 & 0 & 0 & 0 & 0 \end{array}\right],$$
(13)$$B_{e}=\left[\begin{array}{cc} \frac{1}{K_{l}} & 0\\ 0 & \frac{1}{K_{l}}\\ 0 & 0\\ 0 & 0\\ 0 & 0 \end{array}\right],$$
and
(14)$$y\left(t\right)={\left[i_{s\alpha }\left(t\right)\;i_{s\beta }\left(t\right)\right]}^{T},$$
(15)$$C_{e}=\left[\begin{array}{ccccc} 1 & 0 & 0 & 0 & 0\\ 0 & 1 & 0 & 0 & 0 \end{array}\right].$$
Because this extended IM model is nonlinear, the EKF algorithm is used in order to estimate the rotor speed.

## 3. EKF Algorithm for Rotor Speed Estimation

To estimate the rotor speed in real time with the EKF, the continuous-time state equations developed previously in ([Disp-formula FD8a-sensors-24-01744]) and ([Disp-formula FD8b-sensors-24-01744]) are discretized with a sampling period $T_{s}$, using the Euler approximation method [28] as follows.
(16a)$${\hat{x}}_{k+1}=F{\hat{x}}_{k}+Gu_{k},$$
(16b)$${\hat{y}}_{k}=H{\hat{x}}_{k},$$
where
$$\begin{array}{c} {\hat{x}}_{k}={\left[i_{s\alpha ,k}\;i_{s\beta ,k}\;\lambda_{r\alpha ,k}\;\lambda_{r\beta ,k}\;\omega_{r,k}\right]}^{T},\\ y_{k}={\left[i_{s\alpha ,k}\;i_{s\beta ,k}\right]}^{T},\\ u_{k}={\left[v_{s\alpha ,k}\;v_{s\beta ,k}\right]}^{T}, \end{array}$$and
(17a)$$F=I+A_{e}T_{s},$$
(17b)$$G=T_{s}B_{e},$$
(17c)$$H=C_{e},$$
with
$$F=\left[\begin{array}{ccccc} 1-\frac{K_{r}}{K_{l}}T_{s} & 0 & \frac{L_{m}R_{r}}{L_{r}^{2}K_{l}}T_{s} & \frac{pL_{m}\omega_{r,k}}{2L_{r}K_{l}}T_{s} & 0\\ 0 & 1-\frac{K_{r}}{K_{l}}T_{s} & -\frac{pL_{m}\omega_{r,k}}{2L_{r}K_{l}}T_{s} & \frac{L_{m}R_{r}}{L_{r}^{2}K_{l}}T_{s} & 0\\ \frac{L_{m}}{\tau_{r}}Ts & 0 & 1-\frac{1}{\tau_{r}}T_{s} & -\frac{p}{2}\omega_{r,k}T_{s} & 0\\ 0 & \frac{L_{m}}{\tau_{r}}Ts & \frac{p}{2}\omega_{r,k}T_{s} & 1-\frac{1}{\tau_{r}}T_{s} & 0\\ 0 & 0 & 0 & 0 & 1 \end{array}\right],$$
$$G=\left[\begin{array}{cc} \frac{T_{s}}{K_{l}} & 0\\ 0 & \frac{T_{s}}{K_{l}}\\ 0 & 0\\ 0 & 0\\ 0 & 0 \end{array}\right],$$
$$H=\left[\begin{array}{ccccc} 1 & 0 & 0 & 0 & 0\\ 0 & 1 & 0 & 0 & 0 \end{array}\right].$$

In order to take into account the uncertainties and discrepancies between the mathematical model and the real system being observed, the process noise $w_{k}$ and measurement noise $v_{k}$ are introduced into the state-space representation as
(18a)$${\hat{x}}_{k+1}=F{\hat{x}}_{k}+Gu_{k}+w_{k},$$
(18b)$$y_{k}=H{\hat{x}}_{k}+v_{k}.$$
The covariance matrices of the process noise $w_{k}$ and measurement noise $v_{k}$ can be defined as
(19)$$E[\left[\begin{array}{c} v_{i}\\ w_{i} \end{array}\right]\left[\begin{array}{cc} v_{j}^{T} & w_{j}^{T} \end{array}\right]]=\left[\begin{array}{cc} R & S\\ S^{T} & Q \end{array}\right]\delta_{ij},$$
where $\delta_{ij}$ is the Kronecker delta function.

Using conventional equations of the EKF algorithm [29], the induction motor speed is estimated as follows
(20a)$${\hat{x}}_{k+1}^{-}=F{\hat{x}}_{k}^{+}+Gu_{k},$$
(20b)$$P_{k+1}^{-}=J_{k+1}P_{k}^{+}J_{k+1}^{T}+Q_{k},$$
(20c)$$K_{k}=P_{k}^{-}H_{k}^{T}{\left(H_{k}P_{k}^{-}H_{k}^{T}+R_{k}\right)}^{-1},$$
(20d)$${\hat{x}}_{k+1}^{+}={\hat{x}}_{k+1}^{-}+K_{k+1}\left(y_{k+1}-{\hat{y}}_{k+1}\right),$$
(21)$$P_{k+1}^{+}=\left(I-K_{k+1}H_{k+1}\right)P_{k+1}^{-},$$
where the notations − and + stand for before and after the new measurements, $P_{k}$ is the error covariance matrix, $K_{k}$ is the Kalman filter gain, ${\hat{y}}_{k}$ is the estimated output, and $J_{k}$ is the Jacobian matrix used in the EKF to handle nonlinearities, defined as
(22)$$J_{k}=\frac{\partial F}{\partial x}{|}_{x={\hat{x}}_{k}^{+}},$$
with
$$J_{k}=\left[\begin{array}{ccccc} 1-\frac{K_{r}}{K_{l}}T_{s} & 0 & \frac{L_{m}R_{r}}{L_{r}^{2}K_{l}}T_{s} & \frac{pL_{m}\omega_{r}}{2L_{r}K_{l}}T_{s} & \frac{pL_{m}}{2L_{r}K_{l}}T_{s}\lambda_{\beta r}\\ 0 & 1-\frac{K_{r}}{K_{l}}T_{s} & -\frac{pL_{m}\omega_{r}}{2L_{r}K_{l}}T_{s} & \frac{L_{m}R_{r}}{L_{r}^{2}K_{l}}T_{s} & -\frac{pL_{m}}{2L_{r}K_{l}}T_{s}\lambda_{\alpha r}\\ \frac{L_{m}}{\tau_{r}}T_{s} & 0 & 1-\frac{1}{\tau_{r}}T_{s} & -\frac{p}{2}\omega_{r}T_{s} & -\frac{p}{2}T_{s}\lambda_{\beta r}\\ 0 & \frac{L_{m}}{\tau_{r}}T_{s} & \frac{p}{2}\omega_{r}T_{s} & 1-\frac{1}{\tau_{r}}T_{s} & -\frac{p}{2}T_{s}\lambda_{\alpha r}\\ 0 & 0 & 0 & 0 & 1 \end{array}\right].$$

## 4. Noise Covariance Matrices Estimation with a Modified Subspace Model Identification Approach

As mentioned earlier, using the Kalman filter requires defining the process and measurements covariance matrices. These matrices are essential for quantifying the noise levels and model uncertainties. Gererally, the parameters of the noise covariance matrices *Q* and *R* are determined by a trial-and-error-process that can be challenging. In fact, improper or poorly calibrated values of *Q* and *R* can lead to inaccurate estimations of the system states.

In order to relieve this inconvenience, the noise covariance matrices can be estimated. Among all of the tuning techniques available in the literature, specific attention is paid herein to the noise covariance matrices estimation with a modified subspace model identification approach [20].

The solution proposed in this approach consists of
Identifying state-space matrices $A_{d}$, $B_{d}$, $C_{d}$, and $D_{d}$ and state sequence ${\hat{X}}_{f,N}$ with $f,N\in N_{*}^{+}$ from the available input–output data using the subspace model identification method. The identified state sequence ${\hat{X}}_{f,N}$ can be defined as
(23)$${\hat{X}}_{f,N}=\left[\begin{array}{c} {\hat{X}}_{f}\\ {\hat{X}}_{f+1}\\ \vdots \\ {\hat{X}}_{f+N-1} \end{array}\right]\in R^{n_{x}\times N},$$
with ${\hat{X}}_{f,N}\in R^{n_{x}}$;Comparing the identified state-space model with the deterministic part of the model used in the Kalman filter. To this end, both models have to have the same basis. Therefore, we enact a basis change using the transformation matrix *T*, which can be computed as
(24)$$\Gamma_{f}\left(F,H\right)T={\hat{\Gamma }}_{f}\left(A_{d},C_{d}\right),$$
where $\Gamma_{f}\left(F,H\right)$ is the observability matrix of the model used in the Kalman filter defined as
(25)$$\Gamma_{l}\left(F,H\right)={\left[H^{T}\;{\left(HF\right)}^{T}\cdots {\left(HF^{l-1}\right)}^{T}\right]}^{T},$$
and ${\hat{\Gamma }}_{f}\left(A_{d},C_{d}\right)$ is the observability matrix of the identified model using the subspace model identification method defined as
(26)$${\hat{\Gamma }}_{l}\left(A_{d},C_{d}\right)={\left[C_{d}^{T}\;{\left(C_{d}A_{d}\right)}^{T}\cdots {\left(C_{d}A_{d}^{l-1}\right)}^{T}\right]}^{T}.$$Once $\hat{T}$ is estimated with a Moore Penrose pseudo inverse, the state sequence ${\hat{X}}_{f,N}$ can be moved into the “good” state basis as follows
(27)$${\hat{\hat{X}}}_{f,N}=\hat{T}{\hat{X}}_{f,N}.$$Computing the residuals as
(28)$$\left[\begin{array}{c} {\hat{Q}}_{f,1,N-1}\\ {\hat{R}}_{f,1,N-1} \end{array}\right]=\left[\begin{array}{c} {\hat{\hat{X}}}_{f+1,N}\\ Y_{f,1,N-1} \end{array}\right]-\left[\begin{array}{cc} F & G\\ H & 0 \end{array}\right]\left[\begin{array}{c} {\hat{\hat{X}}}_{f,N-1}\\ U_{f,1,N-1} \end{array}\right],$$
where $\hat{\hat{X}}$ with $f,N\in N_{*}^{+}$ represents the state sequence estimate in the “good” state basis performed with the subspace model identification method, and ${\hat{Q}}_{f,1,N-1}$ and ${\hat{R}}_{f,1,N-1}$ are residuals used to estimate the covariance matrices *Q* and *R*.Transforming these discrepancy measurements into covariance matrix estimates. This part will be detailed next.

### 4.1. Subspace Model Identification

According to [30], consider the minimal system
(29a)$$x_{k+1}=A_{d}x_{k}+B_{d}u_{k}+Ke_{k},$$
(29b)$$y_{k}=C_{d}x_{k}+D_{d}u_{k}+e_{k},$$
with $e_{k}$ as a white-noise sequence that is uncorrelated with $u_{k}$.

By taking the instrumental-variable matrix $Z_{N}$ equal to
(30)$$Z_{N}=\left[\begin{array}{c} U_{0,l,N}\\ Y_{0,l,N} \end{array}\right],$$
where $U_{0,l,N}$ and $Y_{0,l,N}$ with $l,N\in N_{*}^{+}$ are the block Hankel matrices constructed from the input–output data, and by considering the following least-squares problem
(31)$$\left[\begin{array}{cc} {\hat{L}}_{N}^{u} & {\hat{L}}_{N}^{z} \end{array}\right]=\arg\min_{L^{u},Lz}||Y_{l,l,N}-\left[\begin{array}{cc} L^{u} & L^{z} \end{array}\right]\left[\begin{array}{c} U_{l,l,N}\\ Z_{N} \end{array}\right]|{|}_{F}^{2},$$
which is solved by a QR factorization
(32)$$\left[\begin{array}{c} U_{f,l,N}\\ U_{0,l,N}\\ Y_{0,l,N}\\ Y_{f,l,N} \end{array}\right]=\left[\begin{array}{ccc} R_{11} & 0 & 0\\ R_{21} & R_{22} & 0\\ R_{31} & R_{32} & R_{33} \end{array}\right]\left[\begin{array}{c} Q_{1}\\ Q_{2}\\ Q_{3} \end{array}\right],$$
it can be shown [30] that
(33)$$R_{32}R_{22}^{-1}\left[\begin{array}{c} U_{0,l,N}\\ Y_{0,l,N} \end{array}\right]=\Gamma_{f}\left(A_{d},C_{d}\right){\hat{X}}_{f,N}.$$
Via the following singular value decomposition
(34)$$R_{32}R_{22}^{-1}\left[\begin{array}{c} U_{0,l,N}\\ Y_{0,l,N} \end{array}\right]={\upsilon \Sigma \nu }^{T},$$
we obtain an estimate of the observability matrix as follows
(35)$${\hat{\Gamma }}_{f}\left(A_{d},C_{d}\right)=\upsilon \Sigma^{1/2},$$
whereas
(36)$${\hat{X}}_{f,N}=\Sigma^{1/2}\nu^{T}.$$

As mentioned earlier, the state sequence ${\hat{X}}_{f,N}$ has to be in the “good” state basis corresponding to the state-space realization used in the Kalman filter (*F*, *G*, *H*). Thus, we enact a basis change using Equation (Equation 24).

Knowing the estimated state sequence ${\hat{\hat{X}}}_{f,N}$, we can quantify the discrepancy between this prior information and the model used in the Kalman filter using Equation (Equation 28).

### 4.2. Noise Covariance Matrices Estimation

We can now determine accurate estimates of the covariance matrices *Q* and *R* using the residuals computed in Equation (Equation 28), as follows [21].
(37)$$\left[\begin{array}{cc} \hat{R} & \hat{S}\\ {\hat{S}}^{T} & \hat{Q} \end{array}\right]=\lim_{N\to \infty }\frac{1}{N}\left[\begin{array}{c} {\hat{Q}}_{f,1,N-1}\\ {\hat{R}}_{f,1,N-1} \end{array}\right]\left[\begin{array}{cc} {\hat{Q}}_{f,1,N-1}^{T} & {\hat{R}}_{f,1,N-1}^{T} \end{array}\right].$$

## 5. Induction Motor Speed Estimation with Noise Covariance Matrices Estimation

As discussed earlier, the speed induction motor estimation using EKF requires a dynamic state space model, as presented in Equations ([Disp-formula FD8a-sensors-24-01744]) and ([Disp-formula FD8b-sensors-24-01744]). This dynamic model, which includes the rotor speed as the state space, is non-observable. Alternatively, the noise covariance matrices estimation with the modified subspace model identification approach presented previously requires an observable model.

As a solution, we propose to identify only the dynamic model, which does not include the rotor speed as the state variable, as described by Equations ([Disp-formula FD1a-sensors-24-01744]) and ([Disp-formula FD1b-sensors-24-01744]). This fourth-order model leads to the determination of the covariance matrices $Q_{1}$ and $R_{1}$ of the system noise and measurement noise, respectively.

In order to determine the covariance matrices of the dynamic model, which is able to estimate the rotor speed, the measurement noise covariance matrix *R* is the same as the fourth-order model beacause it depends only on the measurements, unlike the system noise covariance matrix, which depends on the estimated states.

Thus, in the new system noise covariance matrix, we only have to adjust the parameter μ as shown below.
(38)$$Q=\left[\begin{array}{ccccc} q_{11} & q_{12} & q_{13} & q_{14} & 0\\ q_{21} & q_{22} & q_{23} & q_{24} & 0\\ q_{31} & q_{32} & q_{33} & q_{34} & 0\\ q_{41} & q_{42} & q_{43} & q_{44} & 0\\ 0 & 0 & 0 & 0 & \mu \end{array}\right],$$
where the parameters $q_{ny}$ with *n* number of rows and *y* number of columns are determined with a modified subspace model identification approach.

## 6. Results and Discussion

To justify the need for the automated tuning of the EKF, the speed estimation algorithm of the extended Kalman filter tuned by a trial-and-error-process was tested on our test bench, as presented below.

### 6.1. Experimental Setup

The experimental test setup designed to assess the EKF algorithm was composed of a squirrel-cage induction motor, with 4 kW, 220 V, 50 Hz, and two poles. This motor was controlled by an AC drive, coupled to a permanent magnet synchronous generator (PMSG) operating as a generator and producing a resisting torque, thus representing the load (Figure 2).

The induction motor parameters were given by the manufacturer as
$$R_{s}=1.47\Omega ;\;R_{r}=0.78\Omega ;\;L_{1s}=5.16H;$$
$$L_{1r}=0;\;R_{m}=917.71\Omega ;\;L_{m}=90.139H.$$

To validate the accurate estimation of the rotor speed using the EKF algorithm, an encoder of 1024 points placed at the end of the machine shaft provided the rotor position at each sampling instant, allowing the measurement of the real-time motor speed. In addition, the system integrated sensors to measure the phase voltages and currents, which represent the inputs and outputs of the motor, respectively. These sensor measurements were crucial for providing the necessary data inputs to the EKF algorithm to iteratively refine and enhance the estimation of the rotor speed. The hardware infrastructure supporting this setup included a DS1104 board as an interface between the physical components of the system and the simulation environment in MATLAB/Simulink2023a. The sampling period was $Ts=10^{-3}$ (s).

### 6.2. Experimental Results

The covariance matrices used in the EKF algorithm, tuned by a trial-and-error method until satisfactory estimation performance was obtained, were
(39)$$Q=\left[\begin{array}{ccccc} \lambda & 0 & 0 & 0 & 0\\ 0 & \lambda & 0 & 0 & 0\\ 0 & 0 & \lambda & 0 & 0\\ 0 & 0 & 0 & \lambda & 0\\ 0 & 0 & 0 & 0 & \mu \end{array}\right],$$
with λ=2 and μ=20,
(40)$$R=\left[\begin{array}{cc} 10^{-3} & 0\\ 0 & 10^{-3} \end{array}\right],$$
and the error covariance matrix was initialized as
(41)$$P=\left[\begin{array}{ccccc} 1 & 0 & 0 & 0 & 0\\ 0 & 1 & 0 & 0 & 0\\ 0 & 0 & 1 & 0 & 0\\ 0 & 0 & 0 & 1 & 0\\ 0 & 0 & 0 & 0 & 1 \end{array}\right].$$

Figure 3 and Figure 4 show the estimated currents $is_{\alpha }$ and $is_{\beta }$, respectively, using the trial-and-error method, whereas Figure 5 shows the estimated speed of the EKF using the trial-and-error method.

As we can see in the above figure, the estimated speed was very noisy. This can be justified by the μ coefficient, which was very large.

In order to minimize the effect of the noise in the estimated speed, the coefficient μ had to be reduced. However, a smaller value of μ led to the inaccurate estimation of the motor speed. In fact, the process noise covariance matrix *Q* represents the uncertainty in the system dynamics and reflects the model accuracy in predicting the state evolution of the system. The values of the elements in the process noise covariance matrix determine how much the filter trusts the predicted state versus the measured state. Higher values indicate higher uncertainty in the system, leading to more reliance on the measurements. Furthermore, the error between the estimated and measured speed was more significant in the time interval of 80≤t≥85 s.

A second attempt was made to estimate the speed with the same noise covariance matrices used in the first test (Figure 6, Figure 7 and Figure 8).

As shown in Figure 8, the speed estimation using the same covariance matrices was inaccurate in the time interval of 50≤t≥70 s. Therefore, the noise matrices must be modified each time.

**Figure 8 sensors-24-01744-f008:**
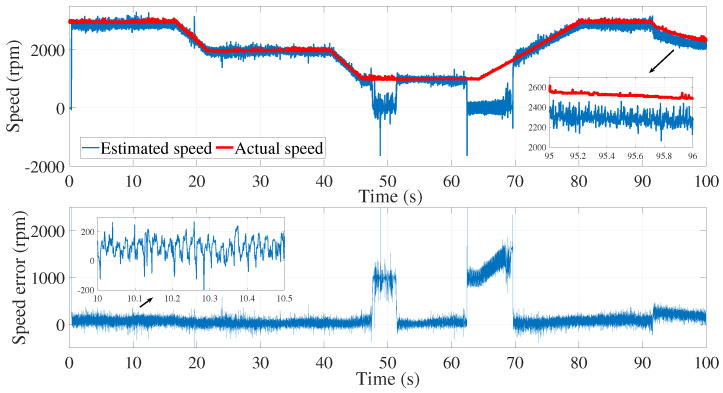
Motor speed estimation using EKF tuned by a trial and error method—second test.

Consequently, manual tuning of the EKF using trial-and-error-process was proven to be inefficient. It is time-consuming and requires significant effort from an experienced operator. Moreover, it involves the configuration of multiple parameters of the noise covariance matrices.

To overcome this difficulty, the noise covariance matrices were determined using a subspace model identification approach.

Figure 9 gives the block diagram of the experimental setup used to estimate these noise covariance matrices.

As shown in the above block diagram, a speed setpoint was provided. Since the model used for estimating the rotor speed is nonlinear, and the subspace model identification method requires a linear time-invariant state-space representation, the nonlinear model was linearized around the known nominal rotor speed.

In addition, system identification requires exciting the inputs and outputs of the system (voltages and currents, respectively). To this end and from the same test bench, a pseudo random binary sequence (PRBS) was applied as the desired speed (Figure 10) and varied around the nominal speed (2920 rpm).

Excited voltages and currents were then used to identify a discrete-time linear state space of the system.

The identification efficiency was evaluated by comparing the identified and actual currents (system outputs) as shown in Figure 11 and Figure 12.

The best fits obtained by comparing the identified and actual currents were 78.73% for $i_{s\alpha }$ and 79.73% for $i_{s\beta }$.

In fact, mathematical models are often simplified approximations of reality and may not always capture all the details of the system’s behavior or take into account the nonlinearities.

The noise covariance matrices were then determined using the modified subspace model identification
(42)$$Q_{1}=\left[\begin{array}{cccc} 2.46 & 0.08 & -1.96 & -3.88\\ 0.08 & 0.81 & 2.420 & -1.58\\ -1.96 & 2.42 & 9.97 & -0.87\\ -3.88 & -1.58 & -0.87 & 10.14 \end{array}\right]$$
(43)$$R=\left[\begin{array}{cc} 2.51 & -3.90\\ -3.90 & 12.21 \end{array}\right].$$

We can now reconstruct the process noise covariance matrix as
(44)$$Q=\left[\begin{array}{ccccc} 2.46 & 0.08 & -1.96 & -3.88 & 0\\ 0.08 & 0.81 & 2.420 & -1.58 & 0\\ -1.96 & 2.42 & 9.97 & -0.87 & 0\\ -3.88 & -1.58 & -0.87 & 10.14 & 0\\ 0 & 0 & 0 & 0 & \mu \end{array}\right].\;$$

Using the identified process noise covariance matrix *Q* and measurement noise covariance matrix *R* to estimate the rotor speed of the same test bench described before, and by varying the μ coefficient until a satisfactory result was determined, as in the manual tuning method, the resulting estimations are presented in (Figure 13, Figure 14 and Figure 15).

Accurate speed estimation was obtained for μ=40.

The speed estimation error decreased compared to the trial-and-error process tuning.

We then applied the same noise covariance matrices to the second test (Figure 16, Figure 17 and Figure 18).

As we can see, the identified noise covariance matrices were valid for both tests, unlike the trial-and-error-process method, where the noise covariance matrices had to be set each time.

Furthermore, Figure 19 shows a comparison between the measured speed, the estimated speed using manual tuning, and the estimated speed using the modified subspace identification approach of the first test.

As shown, the effect of noise was minimized using automated tuning of the covariance matrices.

Finally, the table below provides a comparison between the mean square error of the estimated speed for both the trial-and-error process method and the automated tuning of the EKF (Table 1).

The mean square error between the actual rotor speed and the estimated speed is defined as
(45)$$E=\frac{1}{n}\sum_{i=1}^{n}{\left(s_{i}-e_{i}\right)}^{2},$$
where *n* is the number of samples, *s* is the actual speed, and *e* is the estimated speed.

It is observed that the estimation significantly improved for both the first and second test.

### 6.3. Performance Evaluations of Covariance Matrices Identification under Varied Speed and Load Conditions

Previously, we proposed identifying the noise matrices using a linearized model around the nominal speed indicated on the nameplate. However, to test the accuracy of this method in measuring the speeds and its limitations, additional experimental trials involving two scenarios were conducted; the first scenario involved varying the speed far from the nominal point, while the second scenario involved varying the load during operation.

#### 6.3.1. First Scenario

We determined the noise covariance matrices by introducing speed variations far from the nominal point.

The rotor speed of both test 1 and 2, presented previously, was then estimated using these noise covariance matrices.

The identification of the noise covariance matrices worked regardless of the excitation protocol used, whether it was near the nominal speed at 2285 rpm or far from it at 1088 rpm, for both tests 1 and 2 (Figure 20, Figure 21, Figure 22 and Figure 23). However, identifying the covariance matrices around the nominal speed led to better speed estimation.


**
Speed variations around 30% of the nominal speed (1088 rpm).
**


**Figure 20 sensors-24-01744-f020:**
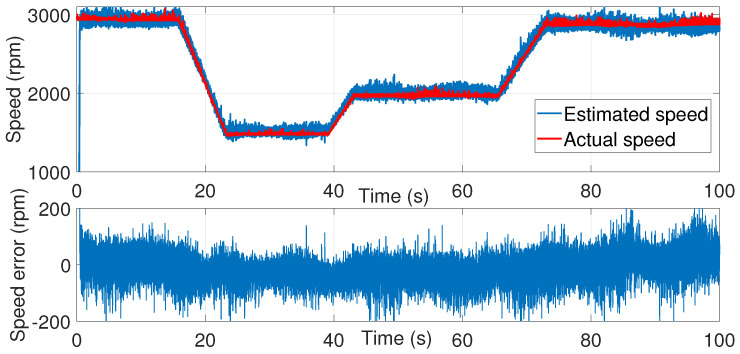
IM speed estimation using identified noise covariance matrices, obtained by speed variations around 1088 rpm—first test.

**Figure 21 sensors-24-01744-f021:**
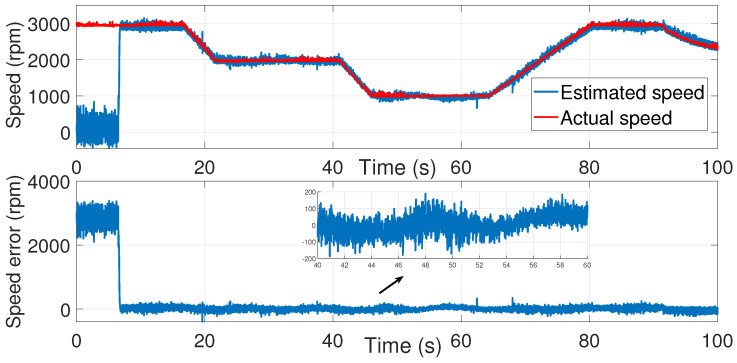
IM speed estimation using identified noise covariance matrices, obtained by speed variations around 1088 rpm—second test.


**
Speed variations around 70% of the nominal speed (2285 rpm).
**


**Figure 22 sensors-24-01744-f022:**
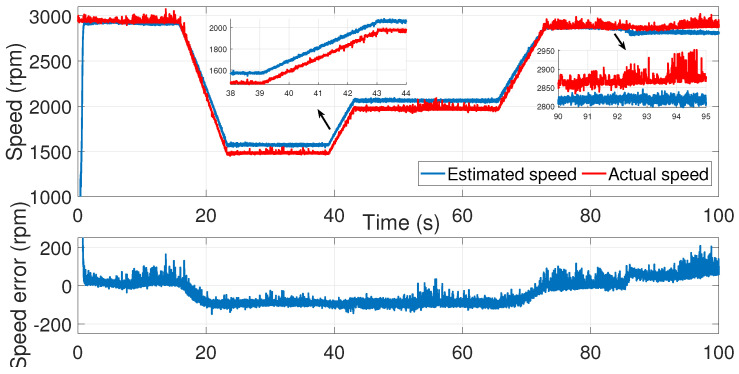
IM speed estimation using identified noise covariance matrices, obtained by speed variations around 2285 rpm—first test.

**Figure 23 sensors-24-01744-f023:**
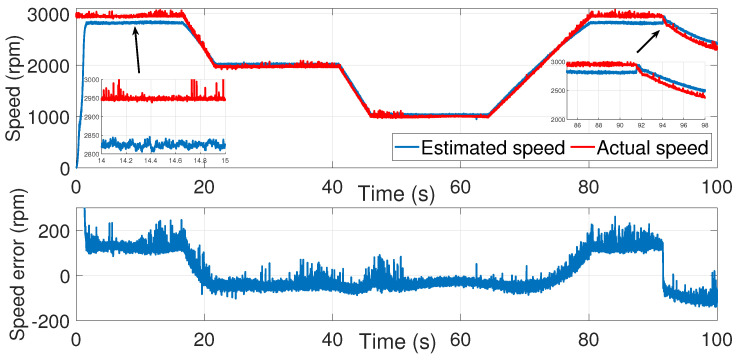
IM speed estimation using identified noise covariance matrices, obtained by speed variations around 2285 rpm—second test.

#### 6.3.2. Second Scenario

The rotor speed of both test 1 and 2 was estimated using the identified noise covariance matrices at the following settings.

Similar to the identification of the covariance matrices using the speed variation far from the nominal speed, identifying covariance matrices using the motor operation at different loads levels yielded poorer results (Figure 24, Figure 25, Figure 26 and Figure 27). This can be explained by the use of the Kalman filter model based on operation at the nominal point. At full load, the system operates closer to its intended performance, leading to more accurate measurements.


**
Operating at 40% of the nominal load.
**


**Figure 24 sensors-24-01744-f024:**
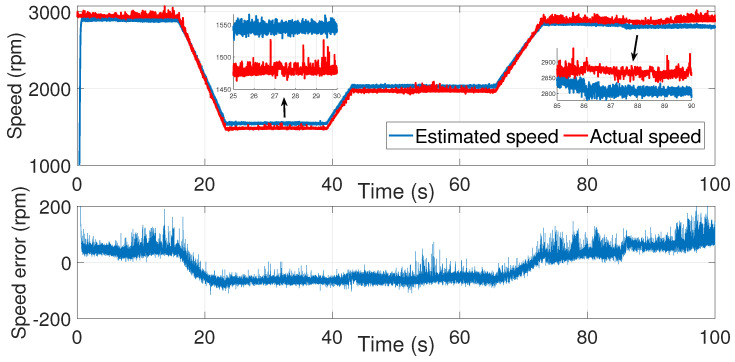
Estimation of the speed using the identified noise covariance matrices, acquired during operation at 40% of the load—first test.

**Figure 25 sensors-24-01744-f025:**
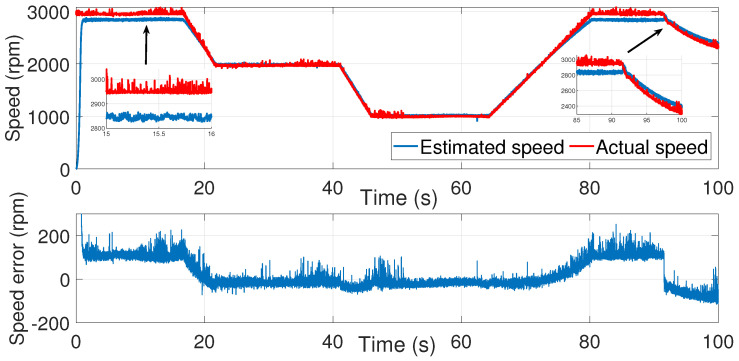
Estimation of the speed using the identified noise covariance matrices, acquired during operation at 40% of the load—second test.


**
Operating at 70% of the nominal load.
**


**Figure 26 sensors-24-01744-f026:**
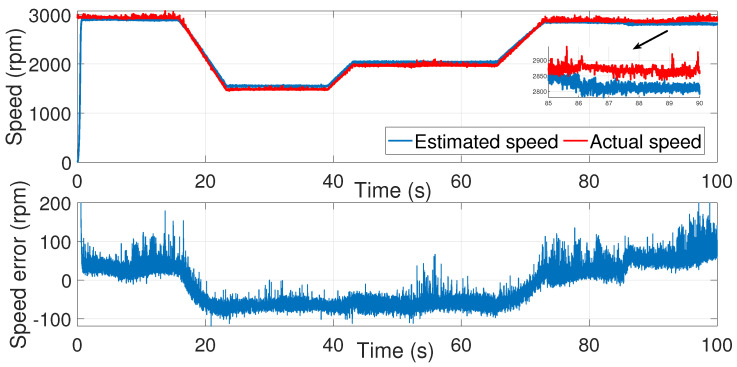
Estimation of the speed using the identified noise covariance matrices, acquired during operation at 70% of the load—first test.

**Figure 27 sensors-24-01744-f027:**
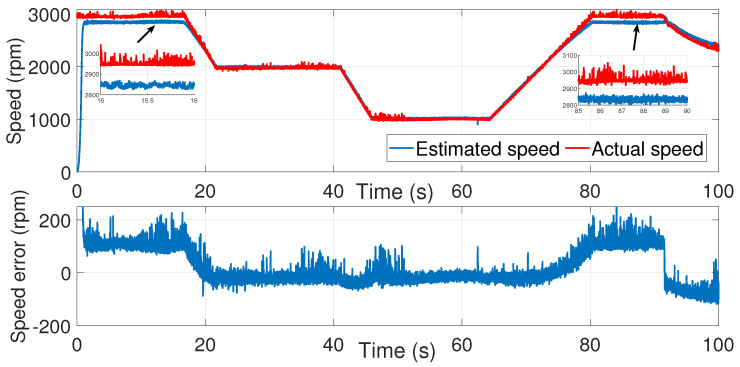
Estimation of the speed using the identified noise covariance matrices, acquired during operation at 70% of the load—second test.

## 7. Conclusions

In this paper, a modified subspace model identification method is used to determine the noise covariance matrices of the EKF in order to estimate the IM rotor speed. The method involves exciting the voltages and currents using PRBS to estimate a discrete state-space model of the IM. This model is then compared to the discrete-time state-space representation involved in the Kalman filter equations. The resulting discrepancy is finally transformed into covariance matrix estimates. Because the subspace state-space system identification requires an observable model, the fourth-order induction motor model is first identified. This model is then used to determine the system noise covariance matrices of the fifth-order model. These estimated matrices are finally used in the EKF algorithm to estimate the rotor speed. The method was tested experimentally, and accurate speed estimation was obtained.

## Figures and Tables

**Figure 1 sensors-24-01744-f001:**
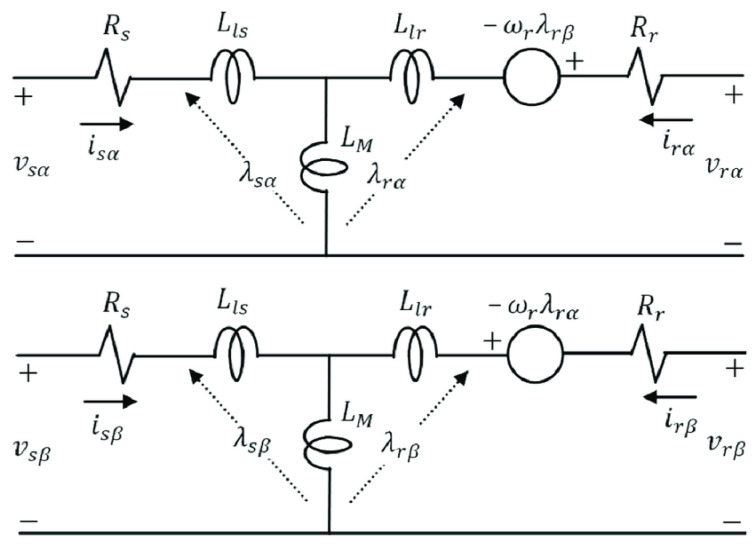
Single-phase equivalent circuit in the stationary reference frame.

**Figure 2 sensors-24-01744-f002:**
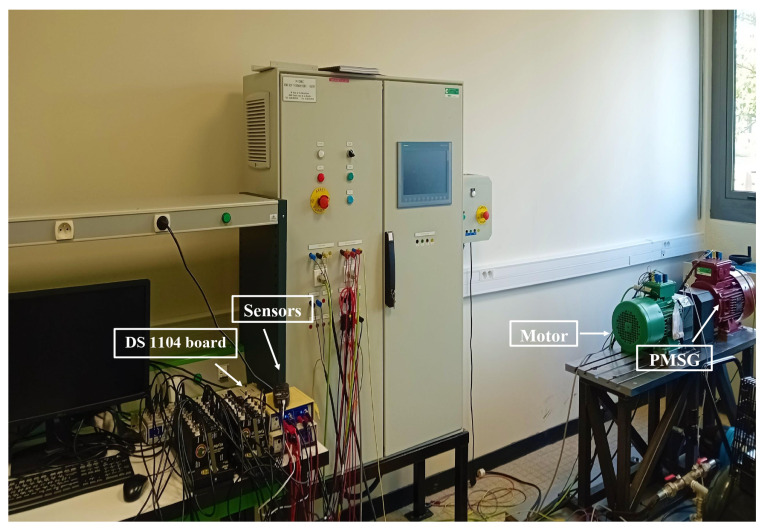
Experimental system.

**Figure 3 sensors-24-01744-f003:**
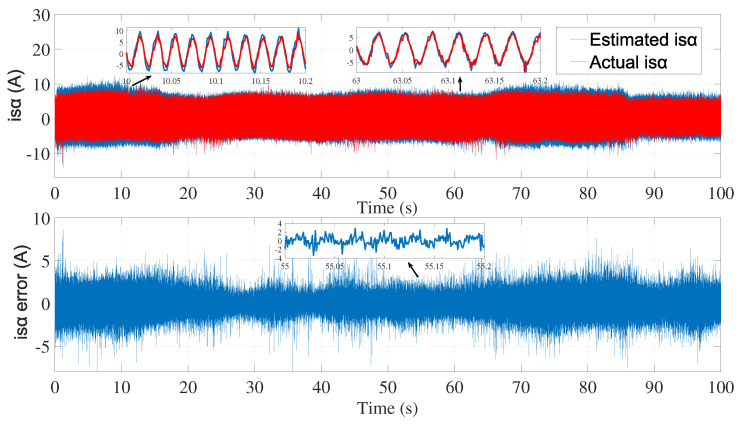
$is_{\alpha }$ estimation using EKF tuned by a trial-and-error method—first test.

**Figure 4 sensors-24-01744-f004:**
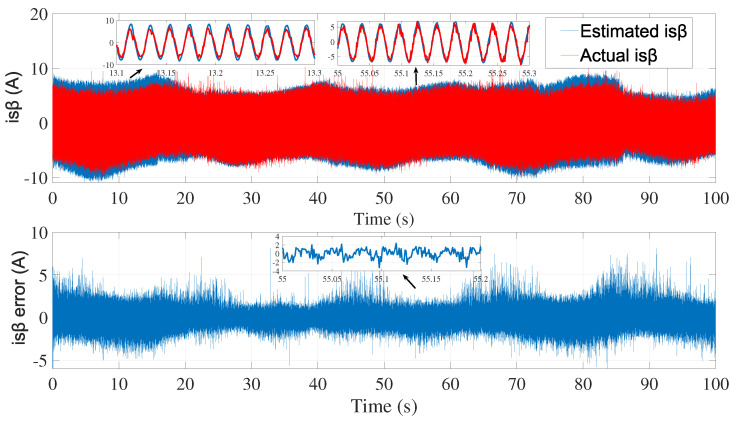
$is_{\beta }$ estimation using EKF tuned by a trial-and-error method—first test.

**Figure 5 sensors-24-01744-f005:**
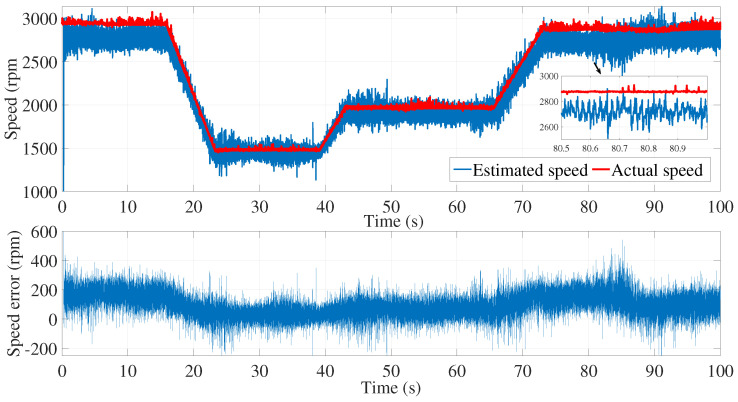
Motor speed estimation using EKF tuned by a trial and error method—first test.

**Figure 6 sensors-24-01744-f006:**
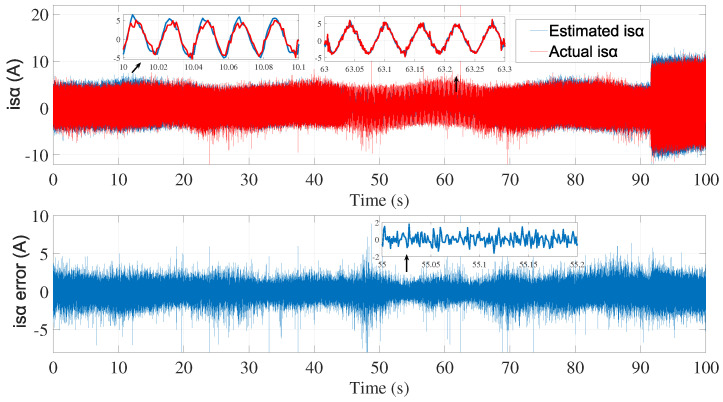
$is_{\alpha }$ estimation using EKF tuned by a trial and error method—second test.

**Figure 7 sensors-24-01744-f007:**
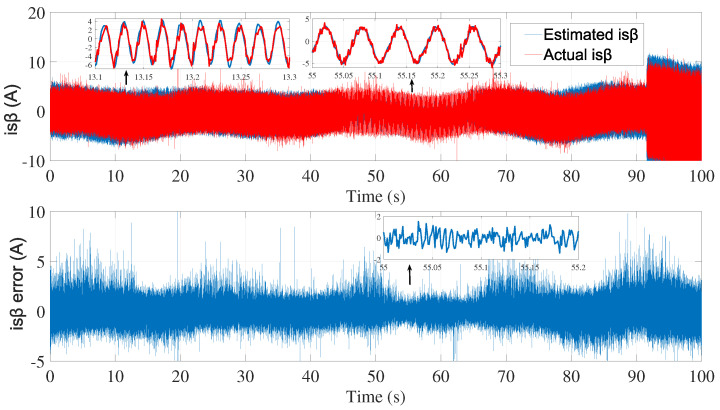
$is_{\beta }$ estimation using EKF tuned by a trial-and-error method—second test.

**Figure 9 sensors-24-01744-f009:**
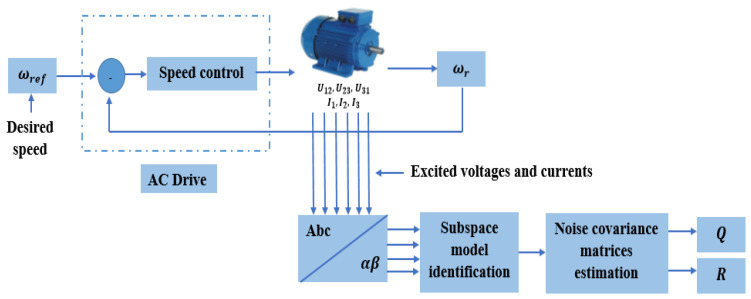
The block diagram of the experimental setup for noise covariance matrices estimation.

**Figure 10 sensors-24-01744-f010:**
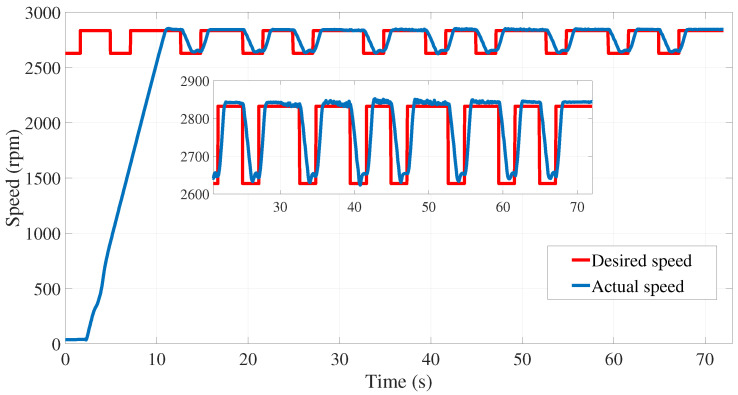
Pseudo random binary sequence exciting the motor voltages and currents used in the subspace model identification.

**Figure 11 sensors-24-01744-f011:**
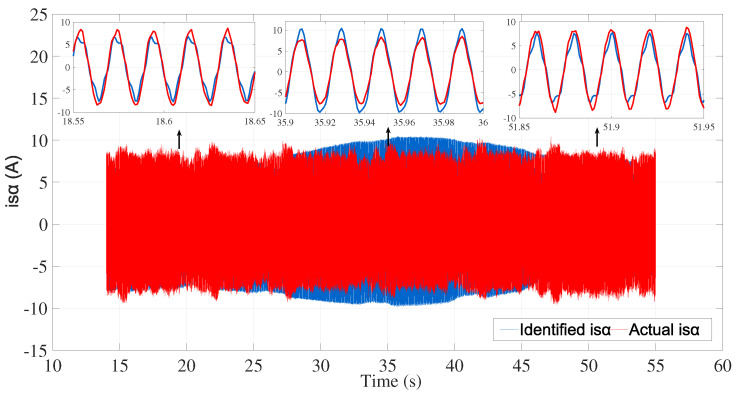
Actual and identified $i_{s\alpha }$.

**Figure 12 sensors-24-01744-f012:**
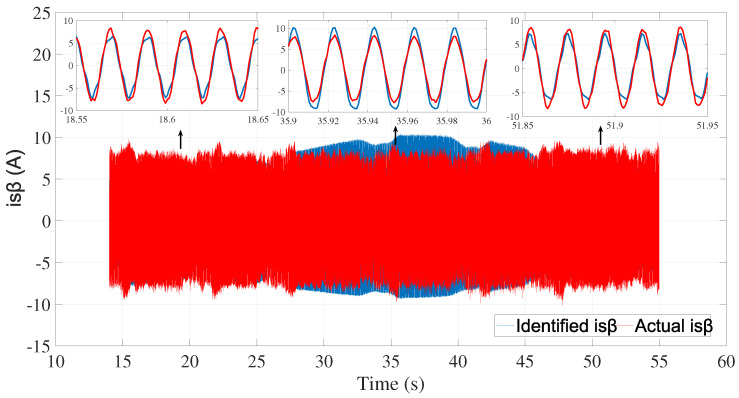
Actual and identified $i_{s\beta }$.

**Figure 13 sensors-24-01744-f013:**
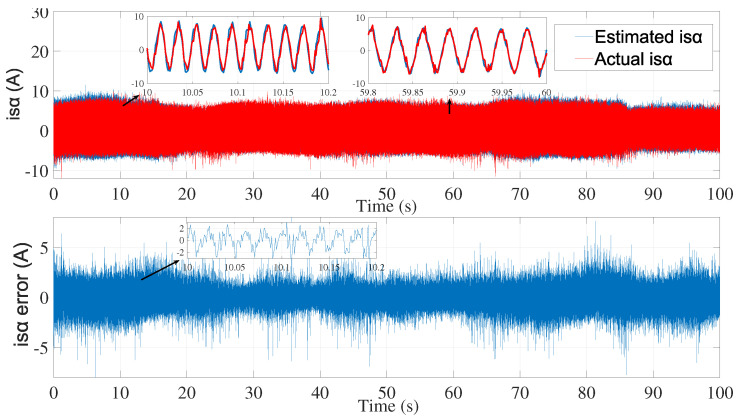
$is_{\alpha }$ estimation with EKF using noise covariance matrices estimation—first test.

**Figure 14 sensors-24-01744-f014:**
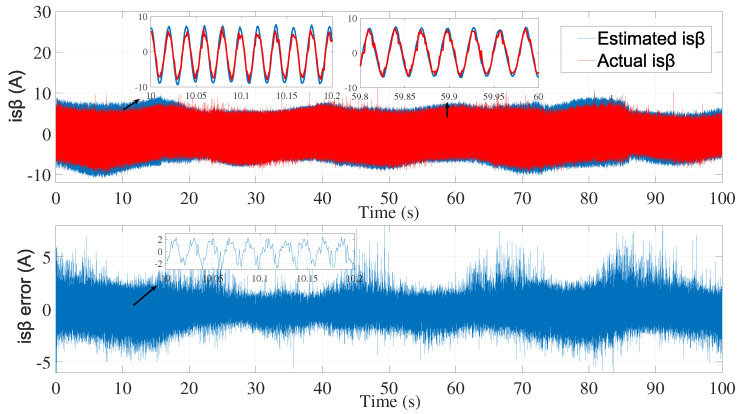
$is_{\beta }$ estimation with EKF using noise covariance matrices estimation—first test.

**Figure 15 sensors-24-01744-f015:**
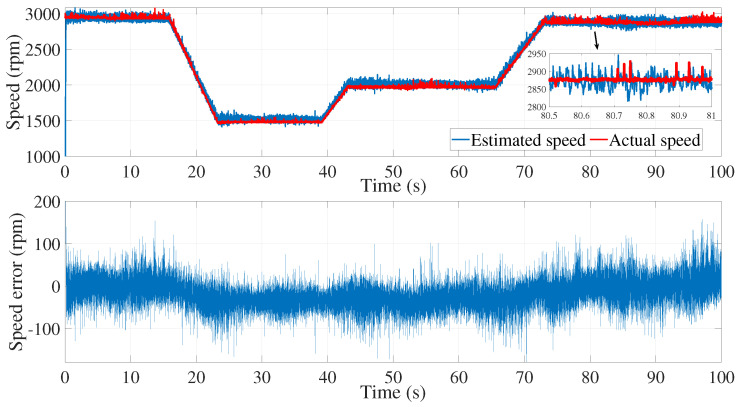
IM speed estimation with EKF using noise covariance matrices estimation—first test.

**Figure 16 sensors-24-01744-f016:**
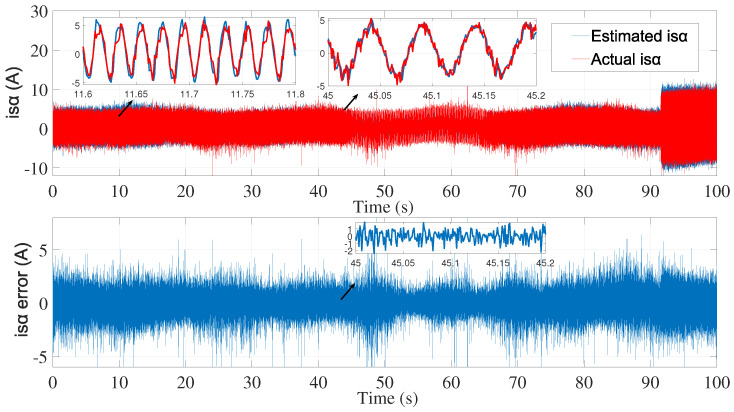
$is_{\alpha }$ estimation with EKF using noise covariance matrices estimation—second test.

**Figure 17 sensors-24-01744-f017:**
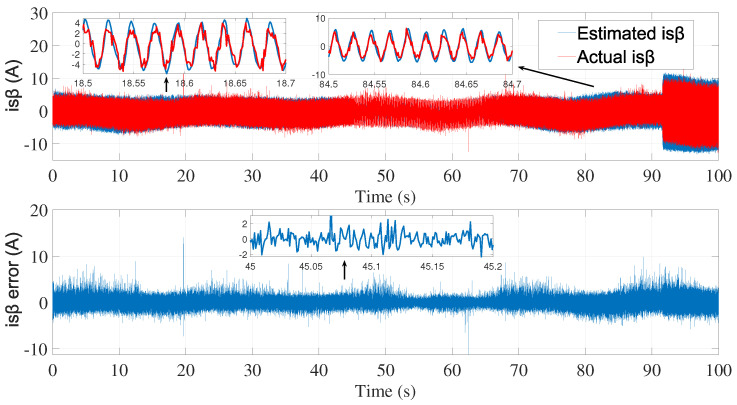
$is_{\beta }$ estimation with EKF using noise covariance matrices estimation—second test.

**Figure 18 sensors-24-01744-f018:**
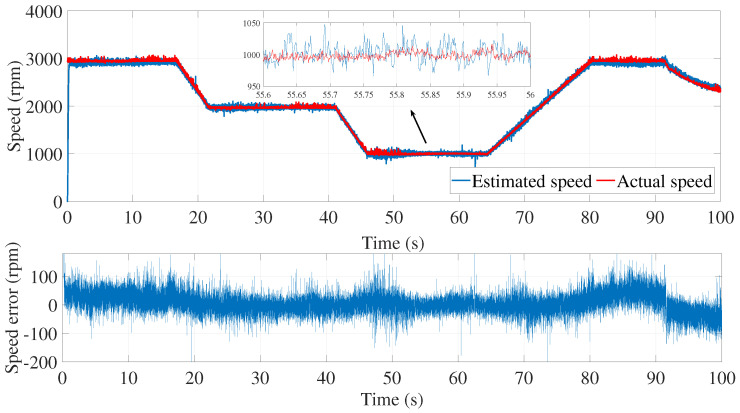
IM speed estimation with EKF using noise covariance matrices estimation—second test.

**Figure 19 sensors-24-01744-f019:**
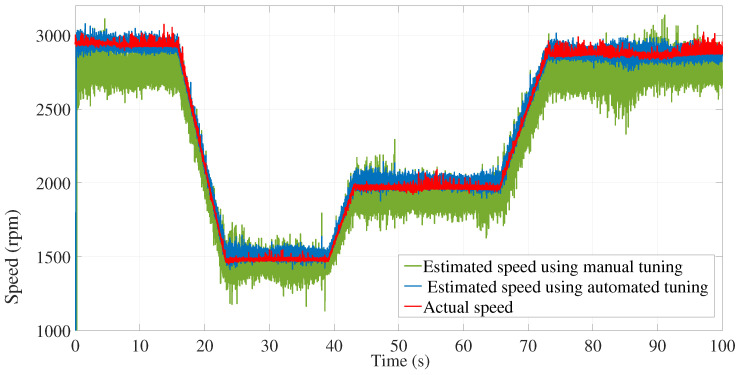
Estimation speed comparison between manual and automated EKF tuning.

**Table 1 sensors-24-01744-t001:** Mean square error of estimated speed.

EKF Tuning	First Test	Second Test
Trial and error process	0.18	0.18
Automated tuning	0.002	0.01

## Data Availability

Data are available upon request from the corresponding author.

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
