# Peer review of "Real-Time Speed Estimation for an Induction Motor: An Automated Tuning of an Extended Kalman Filter Using Voltage–Current Sensors"

_sensors, 2024, doi:10.3390/s24061744_

Round 1
Reviewer 1 Report
Comments and Suggestions for Authors
The authors have proposed a paper on methods for calculating the rotational speed of an asynchronous motor rotor. Currently, attention to more efficient operating modes of systems is most relevant, and calculating engine parameters such as current, voltage and especially speed is a priority. Because the engine is the most common load in the industry. In the introductory part, the authors considered different ways of measuring speed, where they identified the most optimal one – the Kalman filter. In the work, it can be noted that there is an experiment confirming the mathematical analysis carried out in the work. Nevertheless, in my opinion, it would be great to compare this method with different types of load, if possible. It is worth adding more quantitative analysis to the conclusions, yes, this is indicated in the previous section of the discussion, but here it would be possible to emphasize what the accuracy of the measurement is, how much more accurately this method measures speeds, such edits.
In general, the article can be published after minor adjustments.
Reviewer 2 Report
Comments and Suggestions for Authors
This paper focuses on proposing an optimal estimation method for induction motor speed based on extended Kalman filtering. The system estimation model is obtained through the subspace state sequence identification method and compared with the discrete temporal space representation used in the Kalman filter equation to determine the uncertainty of the model, which is expressed as a noise covariance matrix for velocity estimation.
The specific questions are as follows:
1. This article does not analyze the limitations of using the fourth-order model to estimate the noise covariance matrix of the fifth-order model.
2. This article lacks the influence of parameter μ on the overall RPM estimation under fifth-order modeling.
3. The formulas in lines 99-107, 119-132 and 278-288 of this article are not numbered.
4. The inductance units of lines 26-28 are incorrect.
5. This article lacks discussion on how to obtain the actual rotational speed for comparison and error margin.
6. The experimental comparison in this article is not sufficient, and more experiments should be added to test the sensitivity of the proposed rotation speed estimation method to parameters.
7. The "Ts" in line 274 are unitless.
Reviewer 3 Report
Comments and Suggestions for Authors
This manuscript reported the optimal rotor speed estimation for an induction motor using the EKF, which is essential for fault diagnosis in MCSA. The estimated model of a deterministic system derived from available input-output datasets was compared with the discrete-time state-space representation used in the Kalman filter equations, which leads to the determination of model uncertainties. Based on the 5th order nonlinear model, rotor speed is estimated with the optimized EKF algorithm and the used algorithm is tested experimentally.
The content of the manuscript is within the scope of the journal and can be of broad interest to readers. However, in terms of specific content, there is still room for improvement. Therefore, I decided to give the decision of minor revision. It is recommended that the author properly absorb the reviewers' comments and make corresponding improvements and enhancements.
1. For the keywords, 'real-time speed estimation', 'voltage-current sensor', 'fault diagnosis', and 'accuracy' should be added to attract a broader readership.
2. Kalman filter has been extensively used. However, it may perform poorly in the case that the actual model does not coincide with the nominal one. Has its robust issue been considered and how to solve the uncertainty cases? Moreover, if the model perturbations appears, how to realize smoothin? Some latest literature related to Kalman Filter should be introduced briefly (10.1109/TAC.2021.3106861; 10.1016/j.jfranklin.2022.10.050).
3. Table 1 does not adhere to the three-line table format at all.
4. This manuscript lists too many formulas and is too detailed. It is recommended that it be appropriately deleted and only the necessary number of formulas be retained.
5. Page 9, the introduction to the experimental system is too brief, and the necessary parameter configuration needs further introduction and explanation. Otherwise, it is not conducive to repeat such experiments for verification.
Round 2
Reviewer 2 Report
Comments and Suggestions for Authors
There are still some minor editorial issues in this paper, such as the lack of numbering for the equations in lines 312-317, which could be published with minor modification.
Author Response
Dear Reviewer,
We are sorry we missed these equations. We have made the changes.
Best regards.
Comments :
There are still some minor editorial issues in this paper, such as the lack of numbering for the equations in lines 312-317, which could be published with minor modification.